

# Microcanonical and canonical fluctuations in atomic Bose-Einstein condensates – Fock state sampling approach

Maciej Bartłomiej Kruk[1*, 2], Dawid Hryniuk[1, 3], Mick Kristensen[4], Toke Vibel[4], Krzysztof Pawłowski[1], Jan Arlt[4] and Kazimierz Rzążewski[1]

**1** Center for Theoretical Physics, Polish Academy of Sciences,
Al. Lotników 32/46, 02-668 Warsaw, Poland
**2** Institute of Physics, Polish Academy of Sciences,
Al. Lotników 32/46, 02-668 Warsaw, Poland
**3** Department of Physics and Astronomy, University College London,
Gower Street, London, WC1E 6BT, United Kingdom
**4** Center for Complex Quantum Systems, Department of Physics and Astronomy,
Aarhus University, Ny Munkegade 120, DK-8000 Aarhus C, Denmark

⋆ mbkruk@ifpan.edu.pl

## Abstract

The fluctuations of the atom number between a Bose-Einstein condensate and the surrounding thermal gas have been the subject of a long standing theoretical debate. This discussion is centered around the appropriate thermodynamic ensemble to be used for theoretical predictions and the effect of interactions on the observed fluctuations. Here we introduce the so-called Fock state sampling method to solve this classic problem of current experimental interest for weakly interacting gases. A suppression of the predicted peak fluctuations is observed when using a microcanonical with respect to a canonical ensemble. Moreover, interactions lead to a shift of the temperature of peak fluctuations for harmonically trapped gases. The absolute size of the fluctuations furthermore depends on the total number of atoms and the aspect ratio of the trapping potential. Due to the interplay of these effect, there is no universal suppression or enhancement of fluctuations.

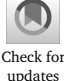

# 1   Introduction

Bose-Einstein condensates are at the heart of current efforts to understand complex many body quantum systems. Typically, investigations of these systems evaluate mean values of a given observable such as the number of atoms in a given state. Further important insights are often offered by higher moments of the relevant probability distribution and thus a full statistical description of such systems would be highly desirable. However, the statistics of complex systems is one of the fundamental problems in many areas of physics and a full description of a gas of bosonic particles - called a Bose gas in this context -is not available.

In this paper, we present a new method to study statistical properties of the ideal and weakly interacting Bose gas at equilibrium. This problem has been studied since the 1940s when E. Schrödinger noticed that the commonly used grand canonical ensemble description of the non-interacting Bose gas leads to absurdly large fluctuations if applied to an isolated system [1]. The questions was further corroborated in the work of R. Ziff, G. Uhlenbeck, M. Kac [2]. They concluded that the canonical fluctuations do not suffer the problem noticed by E. Schrödinger and thus showed that the commonly used statistical ensembles are not equivalent with respect to condensate fluctuations. The statistical problem gained renewed interest [3–10] after 1995, when the first Bose-Einstein condensate in a dilute gas was produced [11, 12]. To reach the necessary temperatures for atomic Bose-Einstein condensation, the gas is isolated as much as possible from its environment. Thus, its statistics should be close to the microcanonical ensemble predictions and it was shown that the microcanonical fluctuations of the three-dimensional (3D) non-interacting gas are expected to be significantly lower than canonical fluctuations [8, 9]. Figure 1 is a sketch that outlines the expected atom number fluctuations of a a non-interacting Bose gas calculated using the statistical ensembles discussed above for the case of $N = 1000$ atoms trapped in a 3D harmonic trap. This clearly shows that the expected fluctuations strongly depend on the chosen ensemble.

For a long time, the problem of condensate fluctuations was an academic one. First observations were performed in an exotic non-isolated condensate made of light [13] showing the grand canonical fluctuations [14]. For atomic condensates, such measurements turned out more difficult since the technical fluctuations of the total number of atoms, due to the noise in the cooling process, were much larger than the equilibrium fluctuations of the condensed

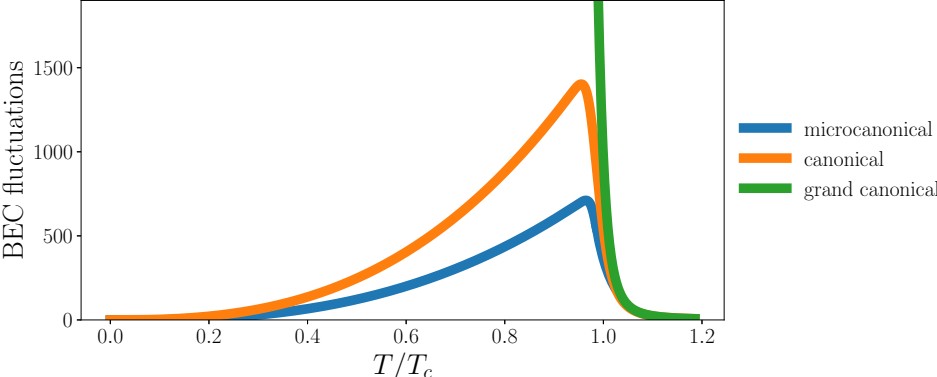

Figure 1: Illustration of the variance of the number of ground state atoms in three statistical ensembles as a function of temperature $T$ in units of the canonical critical temperature $T_c$. When the temperature of an ultracold Bose gas is lowered towards $T_c$ a grand canonical ensemble calculation predicts unphysically large fluctuations (green line). A canonical ensemble calculation does not suffer from this problem and predicts a peak in fluctuations below $T_c$ (orange line). These fluctuations decrease as the number of atoms in the thermal states declines for lower temperatures. A microcanonical calculation (blue line) shows the same features but predicts quantitatively lower peak fluctuations which can be shifted in temperature with respect to the canonical result. This sketch is based on a numerical calculation for $N = 1000$ non-interacting atoms in a three dimensional symmetric harmonic trap.

gas. The situation changed with recent experiments that allowed for unprecedented control of the number of atoms [15, 16]. This enabled the first measurements of the condensate fluctuations [17] and confirmed that the observed fluctuations are closer to the microcanonical predictions [18]. Despite using as many as $5 \times 10^5$ atoms, the experiments are conducted far from the known asymptotic ($N \to \infty$) predictions for the fluctuations [7, 9]. Crucially, for such a large number of atoms, no exact result for microcanonical fluctuations is available.

Moreover, the effect of interactions on condensate fluctuations remains a controversial problem. The only solid results are offered by the Bogoliubov approximation. It has been shown that the fluctuations of an interacting condensate, in the limit of large atom numbers, are up to two times smaller than the ones of the non-interacting gas [19]. The Bogoliubov approximation has been applied for the problem of fluctuations in the canonical [19–24] and then in the microcanonical ensemble [25, 26]. However, this approach only holds for low temperatures and weak interactions. In particular, it does not apply in the vicinity of the critical temperature where the maximal fluctuations of the number of condensed atoms are best determined in the experiment. All other results are either limited to 1D [27] or are sensitive to not very-well-controlled approximations [10, 28]. Various methods give qualitatively different results, as summarized in Fig. 4 of [17]. In the case of 3D interacting systems, methods capable to study statistics for all temperatures exist only to the canonical and the grand canonical ensembles.

In this paper we introduce a new numerical method, called Fock state sampling, to efficiently study the statistical properties of non-interacting and weakly-interacting systems in the canonical ensemble and, in the most restricted, microcanonical ensemble. The method generates a set of representative configurations in a well-chosen configurational space. By appropriate post-selection, we interpolate smoothly between micro- and canonical ensembles. For the non-interacting gas microcanonical results are obtained for as many as $10^5$ atoms in a spherically symmetric trap.

To benchmark our method we first study the Bose gas in a one dimensional box potential with periodic boundary conditions and in a harmonic trap. In these systems, comparisons with exact findings and with the other approximate methods (such as the Bogoliubov and the classical field approximation) are possible. We then proceed to apply our method to the experimentally more relevant three dimensional systems.

The paper is organized as follows. In Sec. 2 we recall the basics of statistical ensembles and formulate our problem. The Fock state sampling method is described in Sec. 3. Section 4 presents our results in one dimensional systems. The results obtained in three dimensional systems are discussed in Sec. 5. This includes the regime between the canonical and the microcanonical ensemble and the interaction-induced shift of the condensate fluctuations in both ensembles. We summarize our findings in Sec. 6.

## 2 Statistical description of a Bose gas

In statistical mechanics the systems of interest at finite temperature are described by statistical ensembles following Gibbs [29]. Such an ensemble is just a collection of copies of the system that fulfills the appropriate conditions. The statistical ensembles are defined by a set of control parameters that correspond to physical constraints imposed on the system. Conceptually, the simplest is the microcanonical ensemble for which the control parameters are: the total number of particles $N$, the total energy $E$, and the volume $V$. For the commonly used harmonic trap this volume is replaced with the trap frequency. This ensemble describes the statistical properties of a fully isolated system, both in terms of exchange of particles and energy. In its quantum version, the properties of the system in a microcanonical ensemble are determined by the partition function $\Gamma(E, N)$, equal to the number of states of $N$ atoms, with total energy $E$. Next in complexity is the canonical ensemble, which also describes a fixed number of particles $N$ but assumes contact with a thermal reservoir imposing a temperature $T$. As a consequence, the energy of the system fluctuates. The least constraining is the grand canonical ensemble which assumes not only a contact with a heat bath but also with a reservoir of particles. The corresponding control parameter $\mu$ is called the chemical potential and plays a role analogous to temperature with respect to particle number. Paradoxically, from the point of view of the computational complexity, the order of ensembles appears inverted, with the grand canonical being the easiest and the microcanonical the most difficult to use.

In this paper, we consider statistical properties of $N$ ultracold trapped bosonic atoms. The Hamiltonian is

$$\hat{H} = \int d^3r \, \hat{\Psi}^\dagger(\boldsymbol{r})\hat{h}\hat{\Psi}(\boldsymbol{r}) + \frac{g}{2} \int d^3r \, \hat{\Psi}^\dagger(\boldsymbol{r})\hat{\Psi}^\dagger(\boldsymbol{r})\hat{\Psi}(\boldsymbol{r})\hat{\Psi}(\boldsymbol{r}), \tag{1}$$

where $\hat{\Psi}(\boldsymbol{r})$ is a bosonic annihilation operator, $\hat{h} = -\frac{\hbar^2\hat{\Delta}}{2m} + V(\boldsymbol{r})$ is the energy density of atoms with mas $m$ placed in a trapping potential $V(\boldsymbol{r})$ and $g$ is a coupling constant related to short range interactions. Here $\hat{\Delta}$ is just the Laplace operator. Note that we consider only contact repulsive interactions, that is $g > 0$.

We consider both, a box potential of length $L$ with periodic boundary condition, i.e. $V = 0$, and a more common harmonic potential $V(x, y, z) = \frac{1}{2}m\left(\omega_\perp^2(x^2 + y^2) + \omega_z^2 z^2\right)$, where we limit our considerations to at most two different frequencies, $\omega_\perp$ and $\omega_z$ responsible for radial and longitudinal confinement, respectively. In the following we refer to $\lambda = \omega_\perp/\omega_z$ as the aspect ratio. In particular, the results for a non-interacting trapped Bose gas do not depend on the explicit values of the trap frequencies, but solely on the parameter $\lambda$. In what follows, whenever we discuss a gas in a harmonic trap we shift the spectrum such that the ground state energy of the non-interacting gas equals zero.

A convenient basis is spanned by the Fock states $|\mathbf{N}\rangle := |N_0, N_1 \ldots\rangle$, where $N_j$ are occupations of the orbitals $\phi_j(\mathbf{r})$ and we choose $\phi_j(\mathbf{r})$ as the eigenstates of a single particle trapped in a potential $V(\mathbf{r})$. In this basis, the field operator $\hat{\Psi}(\mathbf{r})$ is

$$\hat{\Psi}(\mathbf{r}) = \sum_{j=0}^{\infty} \phi_j(\mathbf{r}) \hat{a}_j, \tag{2}$$

where $\hat{a}_j$ ($\hat{a}_j^{\dagger}$) are the bosonic annihilation (creation) operators of atoms in the $j$-th orbital.

The lowest energy state, denoted with $\phi_0(\mathbf{r})$, is the ground state in the non-interacting case. Its occupation $\hat{N}_0 = \hat{a}_0^{\dagger} \hat{a}_0$ fluctuates in all ensembles. The fluctuations are given by

$$\Delta^2 N_0 = \mathrm{Tr}\left\{\hat{N}_0^2 \hat{\rho}\right\} - \left(\mathrm{Tr}\left\{\hat{N}_0 \hat{\rho}\right\}\right)^2, \tag{3}$$

where $\hat{\rho}$ is the density matrix of the gas at equilibrium. The definition of the density matrix depends on the chosen ensembles as given below.

- The microcanonical density matrix is

$$\hat{\rho}_{\mathrm{micro}} = \frac{1}{\Gamma(N, E)} \, \delta\left(\hat{H} - E\right), \tag{4}$$

  where $\delta$ is the Dirac delta function and $\Gamma(N, E)$ is the microcanonical partition function, namely the number of ways to distribute $N$ atoms between energy levels such that the total energy is $E$.

- In the canonical ensemble, the density matrix is defined as

$$\hat{\rho}_{\mathrm{cano}} = \frac{1}{Z(N, \beta)} e^{-\beta \hat{H}}, \tag{5}$$

  where $Z$ is the canonical partition function $Z(N, \beta) := \mathrm{Tr}\left\{e^{-\beta \hat{H}}\right\}$, $\beta := 1/(k_{\mathrm{B}} T)$ and $k_{\mathrm{B}}$ is the Boltzmann constant.

The partition function restricted to the excited states and excited atoms only, is very useful for the computation of the statistics of a condensate. It is denoted with $\Gamma_{\mathrm{ex}}(N_{\mathrm{ex}}, E)$ and $Z_{\mathrm{ex}}(N_{\mathrm{ex}}, \beta)$ in the microcanonical and canonical ensembles, respectively. In particular, $\Gamma_{\mathrm{ex}}(N_{\mathrm{ex}}, E)$ is the number of ways to distribute $N_{\mathrm{ex}}$ atoms between excited energy levels such that the total energy still equals $E$.

In the microcanonical ensemble, the probability that there are $N_0$ atoms in the condensate is thus

$$p_{\mathrm{micro}}(N_0, N, E) := \Gamma_{\mathrm{ex}}(N - N_0, E) / \Gamma(N, E). \tag{6}$$

Analogously, in the canonical ensemble we define

$$Z_{\mathrm{ex}}(N_{ex}, \beta) := \sum_{N_1 + N_2 + \ldots = N_{\mathrm{ex}}} \langle N_0, N_1, \ldots | e^{-\beta \hat{H}} | N_0, N_1, \ldots \rangle, \tag{7}$$

where the sum is over all Fock states with exactly $N_{\mathrm{ex}}$ atoms in all excited energy levels together. The probability of having $N_0$ atoms in the canonical ensemble is then

$$p_{\mathrm{cano}}(N_0, N, \beta) := Z_{\mathrm{ex}}(N - N_0, \beta) / Z(N, \beta). \tag{8}$$

In principle, given the probabilities $p_{\text{micro}}$ and $p_{\text{cano}}$ one could compute the average number of ground state atoms and their fluctuations using

$$\langle N_0 \rangle_{\text{ens}} = \sum_{N_0=0}^{N} p_{\text{ens}}(N_0, N, E)\, N_0\,, \tag{9}$$

$$\left(\Delta^2 N_0\right)_{\text{ens}} = \sum_{N_0=0}^{N} p_{\text{ens}}(N_0, N, E)\, N_0^2 - \langle N_0 \rangle_{\text{ens}}^2\,, \tag{10}$$

where the subscript *ens* indicates the ensemble, labeled *micro* and *cano* in the following.

Computations in the canonical and the microcanonical ensembles are thus conceptually the same. However, while there are efficient methods for calculating the probabilities $p_{\text{cano}}$ in the canonical ensemble [5, 6] the analogous calculations are much more demanding for the microcanonical ensemble. Apart from a few exceptional cases, calculations in the microcanonical ensemble are hence restricted to small systems. Accurate modelling of experiments, therefore, requires the development of computational methods. Below we describe a new numerical method that allows us to perform reliable calculations in both statistical ensembles with up to $10^5$ atoms, without reconstructing the probability distributions $p_{\text{micro}}$ and $p_{\text{cano}}$.

## 3 Fock state sampling method

In practice, the Fock state sampling (FSS) method is a realization of the Metropolis algorithm, widely used in Monte-Carlo simulations [30]. The algorithm generates a Boltzmann-distributed random walk in the space of possible configurations of the system. The set of visited points is interpreted as a collection of microstates of the system representing the canonical ensemble. Importantly, we can also generate points representing the microcanonical ensemble by post-selection from the collection of states. The final collection can be used to compute average values of physical quantities as it is done in the statistical ensembles. Moreover, the method is applicable to weakly interacting Bose gases in the canonical and microcanoncial ensembles as described below.

Our earlier application of the Metropolis algorithm was performed in the framework of the classical field approximation (CFA) [31]. However, in the CFA a fraction of an atom can flow from orbital to orbital due to the lack of discretization. This leads to an unavoidable dependence of the results on the energy cut-off, which is analogous to the ultraviolet catastrophe of the black body radiation prior to M. Planck's introduction of photons.

In our novel FSS method, the algorithm walks among all the Fock states satisfying $\sum N_j = N$. In practice, we also truncate the orbitals at some high value, but now the results are cut-off independent, provided that the cut-off is sufficiently high.

The novelty, apart from the choice of the underlying set of states, rests in the definition of the steps of the random walk. In this respect, the Metropolis algorithm is very flexible - one obtains the correct results provided that the whole space of states is accessible and that the steps satisfy the condition of detailed balance [30]. The FSS method is described in detail below, starting with the non-interacting gas case.

Our walk prescription is physically motivated and quickly provides representative collections of the ensemble copies. In this algorithm every atom has equal chances of jumping away from their occupied mode, that is the probability of a jump from a given orbital is proportional to its occupation. The orbital which the atom jumps to is drawn in proportion to its occupation – a well-known Bose-enhancement factor – plus one, which represents spontaneous transitions. This is analogous to A. Einstein's discussion of $\mathcal{A}$ and $\mathcal{B}$ coefficients for the spontaneous and stimulated emission processes. The probability that an atom jumps from the $j$-th

orbital to the $k$-th orbital is thus proportional[1] to $N_j(N_k + 1)$. The new configuration is accepted and added to the set of copies representing the canonical ensemble if a drawn random number $r \in [0, 1]$ is smaller than $\exp(-\beta(E_\mathrm{f} - E_\mathrm{i}))$, where $E_\mathrm{i}$ and $E_\mathrm{f}$ denote the energy of the Fock state at the beginning of the step and the energy of the candidate Fock state, respectively. A further acceleration of the algorithm is described in appendix A.

Since the initial state can be arbitrary, a number of steps must be made before the random walk begins to represent the ensemble. This initial part of the walk, during which the system "thermalizes" (also referred to as "burn-in" in the Markov Chain Monte Carlo literature), is rejected from the analysis for practical reasons.

We extend this method beyond the non-interacting gas by considering weak contact interactions. In this case, the energy in the Boltzmann factor includes not only the kinetic and potential energy in the trap but also the contribution of the interaction energy. The mean value of the interaction energy, given by the last term in Eq. (1), in the Fock state is given by

$$E_\mathrm{int} = \frac{g}{2} \int \mathrm{d}^3 r \, \langle N_0, N_1 \ldots | \hat{\Psi}^\dagger(\boldsymbol{r}) \hat{\Psi}^\dagger(\boldsymbol{r}) \hat{\Psi}(\boldsymbol{r}) \hat{\Psi}(\boldsymbol{r}) | N_0, N_1 \ldots \rangle \,, \tag{11}$$

and resembles the lowest order perturbation term.

In the case of a harmonic trap, the condensate wave function of the interacting gas differs from the simple Gaussian solution of the single particle ground state. Nonetheless, we only consider the statistics of the population in the lowest orbital in the present version of the method, even in the interacting case. Therefore, we restrict our consideration to very weak interactions. Note however, that for a box with periodic boundary conditions this additional complication does not arise, and $\phi_0(\boldsymbol{r}) = 1/\sqrt{V}$ remains the condensate wave-function even in the presence of interactions.

Once a set of states representing the canonical ensemble is generated we can perform a post-selection routine by choosing a subset that satisfies additional constraints. In particular, one can reduce the set all the way to the microcanonical ensemble by restricting the spread of energies. This is done by introducing a shrinking energy window around the mean energy and removing the states with energies outside this window. In the limiting case where the width of the energy window approaches zero typically only a few states remain. To avoid large statistical error in this case, we determine the microcanonical values by extrapolating the results from wider windows. Further details of this procedure are given in Sec. 5 for an experimentally relevant case.

We extract the important quantities for the discussion of the fluctuations in a Bose-Einstein condensate as follows. The mean number of condensed atoms is given by

$$\bar{N}_0 := \frac{1}{W} \sum_{i=1}^{W} N_{0,i} \,, \tag{12}$$

and its variance is

$$\Delta^2 N_0 := \frac{1}{W} \sum_{i=1}^{W} \left(N_{0,i}\right)^2 - \bar{N}_0^2 \,, \tag{13}$$

where $W$ is the number of configurations, represented in what follows, by Fock states generated by our algorithm. Here $N_{0,i}$ is the occupation of the ground state in the $i$-th copy.

To obtain results for the microcanonical ensemble we use the same formulas (12) and (13), but we restrict the sum to the Fock states that remain after the post-selection, described above and in Sec. 5.

---

[1] Notice that if $j = k$, then the probability is proportional $N_j^2$, i.e. we account for the fact that the atom was first removed and then it returns.

In relation to recent experiments it is furthermore of great interest to evaluate the ratio

$$S := \frac{\max_E \left(\Delta^2 N_0\right)_{\text{micro}}}{\max_T \left(\Delta^2 N_0\right)_{\text{cano}}}, \tag{14}$$

between peak fluctuations in the microcanonical and canonical ensembles. In particular, when $S$ differs significantly from unity, measurements can be used to identify the appropriate ensemble, which recently demonstrated the microcanonical nature of the Bose-Einstein condensation in ultracold gases [18]. Figure 1 illustrates these peak fluctuation in both ensembles.

Importantly, the post-selection process allows for an investigation of the transition from the canonical to microcanonical ensembles. However, it does not give access to the transition from grand canonical to canonical ensembles, which is of interest for photonic condensates [13] and will be the topic of future research.

# 4 Fluctuations in 1D Bose gases

In a first step we apply the Fock state sampling method in a one dimensional setting. This has the advantage that a number of available solutions allow for a more comprehensive benchmarking of the numerical method than the three dimensional case. In particular, we analyze two distinct trapping geometries in 1D. First a ring trap corresponding to a 1D box with periodic boundary conditions is analysed and then a harmonic trap is discussed. In both cases we compare the fluctuations with and without interactions. After benchmarking our method with the canonical ensemble, we use it to discuss the microcanonical case.

## 4.1 1D box with periodic boundary conditions (ring trap)

Figure 2 shows the fluctuations of a Bose gas in a ring trap obtained from canonical ensemble calculations using several different approaches. This includes results based on exact counting statistics [5], the classical field approximation [31], the Bogoliubov approximation, and the Fock state sampling method presented here.

For the non-interacting gas, our present FSS method, as well as the classical field approximation with a well chosen cut-off [32,33], perfectly reproduce the exact result which is known analytically in this case [33]. Moreover, we also find good agreement with the result based on the Bogoliubov approximation within its range of validity at low temperatures. This validates our method for the case of the non-interacting Bose gas in 1D. Moreover, we employ the post-selection process to the FSS method to evaluate the fluctuation in the microcanonical ensemble. This shows a clear reduction of the fluctuation with respect to the canonical expectation.

We now include weak interactions as shown in Fig. 3. No exact canonical results are available for this case, but at low temperature reliable results can be obtained within the Bogoliubov approximation, valid for low temperatures and low interaction strengths. The results based on the Bogoliubov approximation show the expected suppression of fluctuations at low temperature in good agreement with our FSS method computation and earlier results based on the classical field approximation [33]. However, the FSS results show that this suppression is not general, but that the fluctuations surpass the non-interacting case at higher temperature.

Note that this result is a significant change of paradigm. The reduction of fluctuations due to interactions for a Bose gas in a box was stressed in a number of papers that relied on the Bogoliubov approximation [19]. On the basis of Fig. 3 it is now clear that this prediction is limited to low temperatures only. In particular, Fig. 3 shows that the fluctuations of an

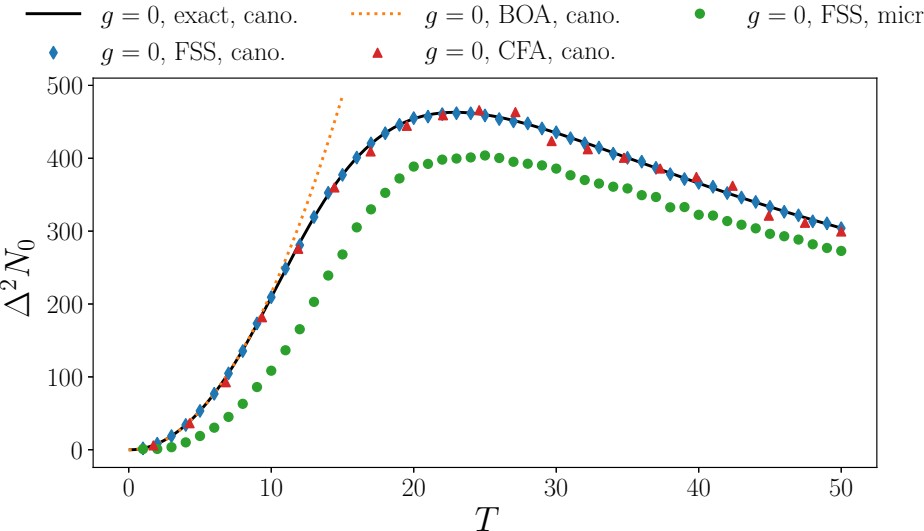

Figure 2: Fluctuations of a non-interacting Bose gas containing $N = 100$ atoms in a 1D ring trap. The variance of $N_0$ as a function of temperature in 1D in canonical ensemble is obtained from several different approaches: FSS method, classical field approximation, Bogoliubov approach (BOA) and an exact method (see text). In addition a microcanonical calculation using the FSS method is presented, showing the clear reduction of the expected fluctuations in this ensemble. The temperature $T$ is given in units $2\pi^2\hbar^2/(mk_B L^2)$.

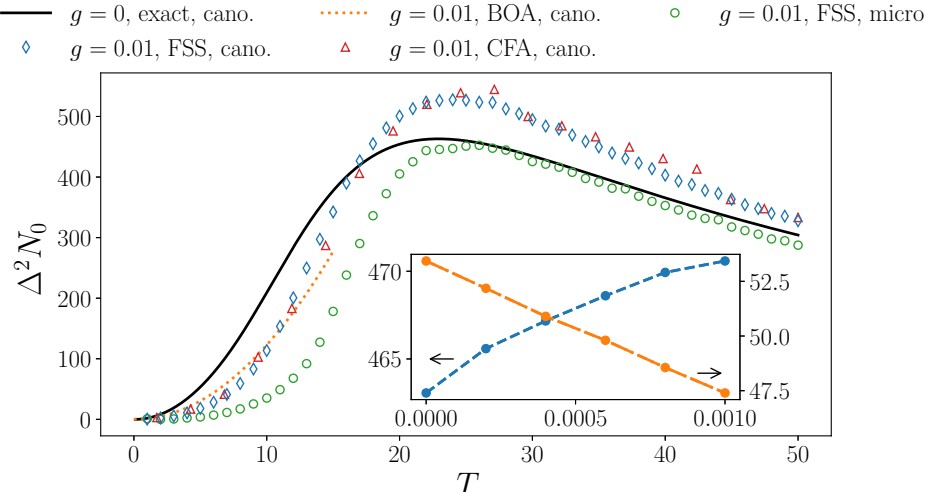

Figure 3: Fluctuations of a weakly-interacting Bose gas containing $N = 100$ atoms in a 1D ring trap. The variance of $N_0$ as a function of temperature is obtained from several different approaches: FSS method, classical field approximation, and Bogoliubov approach. A microcanonical calculation shows a significant suppression of the fluctuations. The exact canonical result from Fig. 2 is shown for comparison. (inset) Variance at a low temperature $T = 5$ (orange, axis on the right) and at the temperature of maximal fluctuations (blue, axis on the left) as a function of the interaction strength $g$, obtained with the FSS method in the canonical ensemble. The arrows indicate the appropriate axis. The interaction strength $g$ and temperature $T$ are given in units $2\hbar^2\pi^2/(mL)$ and $2\pi^2\hbar^2/(mk_B L^2)$ respectively.

interacting gas are larger near their maximal value as our canonical results exceed the non-interacting exact prediction.

This is further corroborated in Fig. 3 (inset). It shows the fluctuations as a function of the interaction strength $g$ in two regimes. At a low temperature $T = 5$ (orange) the fluctuations decrease as a function of interaction strength as predicted within the Bogoliubov approximation. However, at the temperature of the peak fluctuations they increase as a function of interaction strength.

This effect is even more pronounced in a microcanonical calculation, which can be seen by comparing the microcanonical results in Fig. 2 and Fig. 3. Both results are strongly reduced with respect to their corresponding canonical results. In addition the weakly interacting microcanonical result lies considerably below the non-interacting one at low temperatures. At higher temperatures the effect reverses and the weakly-interacting microcanonical variance is larger than non-interacting one . This is evident from a comparison of the green symbols in Fig. 2 and Fig. 3, where the non-interacting canonical result can serve as a guide to the eye.

## 4.2  1D harmonic trap

In a next step we extend our analysis to the experimentally more relevant case of a 1D harmonic trap. For the non-interacting gas, the probability distribution of finding $N_0$ atoms in the ground state in the canonical ensemble is given by

$$p_{\text{cano}}(N_0, N, \beta) = e^{-\beta(N-N_0)\hbar\omega} \prod_{n=N-N_0}^{N} \left(1 - e^{-\beta n \hbar\omega}\right). \tag{15}$$

The exact result for the fluctuations based on this distribution is shown in Fig. 4. Moreover, we show the result based on the Bogoliubov approximation which again agrees with the exact result within its range of validity at low temperatures. Importantly, the FSS method perfectly reproduces the exact result at all temperatures in the 1D harmonic trap.

Based on this agreement, we again include interactions in our analysis. Figure 4 includes the fluctuations of the interacting gas based on the Bogoliubov approximation, which is valid for low temperatures at weak interactions. In clear contrast with the previous case this analysis shows that the fluctuations increase at low temperatures due to the interactions.

The results from our FSS method analysis provide an explanation for this behaviour. At low temperature it agrees well with the results based on the Bogoliubov approximation. However, for higher temperatures the primary effect of interactions is a shift of the temperature of peak fluctuations. This indeed qualitatively corresponds to an expected shift of the critical temperature due to interactions in the system. The peak value of the fluctuations remains almost unchanged and it stays in the range of statistical errors of the FSS method results. Thus the apparent increase of fluctuations at low temperature is primarily caused by the shift of the critical temperature.

Now we turn our attention to the microcanonical ensemble. In this case, there are no closed formulas for the condensate fluctuations, even for the non-interacting gas. Instead one may find the ground state statistics using recurrence relations. Here, one can benefit from the fact that the statistical problem for the 1D harmonic trap is directly related to the classic combinatorial problem of the number of partitions of an integer.

The relevant combinatorial figure in this problem is the number of partitions of an integer $E$ into a sum of $N_{\text{ex}} \leq N$ strictly positive numbers. For the 1D harmonic trap this number is nothing else but $\Gamma_{\text{ex}}(N_{\text{ex}}, E)$ introduced above, where the integer $E$ is the energy expressed in harmonic oscillator units. For example $\Gamma_{\text{ex}}(N_{\text{ex}}, N_{\text{ex}} + 1)$ equals 1, as there is only one way to write the number $N_{\text{ex}} + 1$ as a sum of $N_{\text{ex}}$ positive integers: a single integer equal to 2

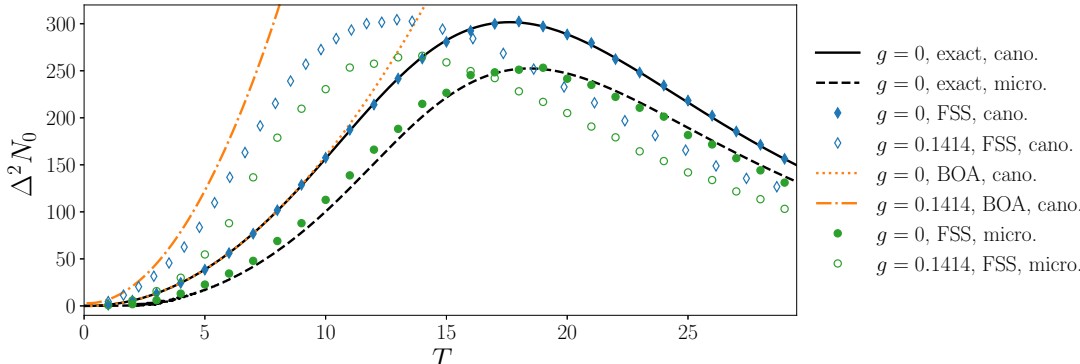

Figure 4: Fluctuations of Bose gas containing $N = 100$ atoms as a function of temperature in a 1D harmonic trap. The fluctuations with and without interactions are computed with several methods: FSS method (symbols), Bogoliubov approach (orange dotted and dot-dashed lines), and exact methods for the non-interacting gas (black solid and dashed lines). Interactions lead to a shift of the temperature of peak fluctuations, resulting in increased fluctuations at low temperature. In addition calculations using a microcanonical (black dashed line, green symbols) ensemble show the significantly lower fluctuations expected in this case. The interaction strength $g$ and temperature $T$ are given in units of $\sqrt{\hbar^3 \omega/m}$ and $\hbar\omega/k_\mathrm{B}$, respectively.

and $N_\mathrm{ex} - 1$ integers equal to 1. In general, the value $\Gamma_\mathrm{ex}(N_\mathrm{ex}, E)$ may be obtained using the recurrence relation

$$\Gamma_\mathrm{ex}(N_\mathrm{ex}, E) = \Gamma_\mathrm{ex}(N_\mathrm{ex}, E - N_\mathrm{ex}) + \Gamma_\mathrm{ex}(N_\mathrm{ex} - 1, E - 1). \tag{16}$$

The first term in this relation, $\Gamma_\mathrm{ex}(N_\mathrm{ex}, E - N_\mathrm{ex})$, is the number of partitions in which the number 1 does not appear. In fact, if we subtract 1 from each element of such a partition, there still remain $N_\mathrm{ex}$ non-zero integers, only that their sum will be $E - N_\mathrm{ex}$. In turn, for every partition of $E$ in which the number 1 appears, one can take this 1 away. Such a new set will contain $N_\mathrm{ex} - 1$ non-zero elements, summing up to $E - 1$. There are $\Gamma_\mathrm{ex}(N_\mathrm{ex} - 1, E - 1)$ such partitions.

The full partition function in the microcanonical ensemble is

$$\Gamma(N, E) = \sum_{N_\mathrm{ex}=0}^{N} \Gamma_\mathrm{ex}(N_\mathrm{ex}, E), \tag{17}$$

and one can easily find the probability distribution of finding $N_0$ atoms in the single particle ground state $p_\mathrm{micro}(N_0, N, E) = \Gamma_\mathrm{ex}(N - N_0, E)/\Gamma(N, E)$. Having $p_\mathrm{micro}(N_0, N, E)$ we invoke formulas (9) and (10) to find the average value and fluctuations of $N_0$.

This exact microcanonical result provides a benchmark for the post-selection process based on the FSS method analysis. As outlined above, the fluctuations in a microcanonical ensemble are calculated by post-selection from a set of states obtained using the FSS method approach in the canonical ensemble, which amounts to a reduction of the energy spread. Figure 4 includes a comparison of our numerical result with the exact calculation. The two are in excellent agreement for the entire temperature range and show that the fluctuations in the microcanonical ensemble are always smaller than in the canonical one.

The post-selection works equally well for interacting systems and Fig. 4 also shows microcanonical the ground state atom number fluctuations in a weakly interacting Bose gas. Similar to the canonical interacting result the maximal fluctuations are shifted to lower temperature (energy) in qualitative agreement with the expected shift of the critical temperature due to

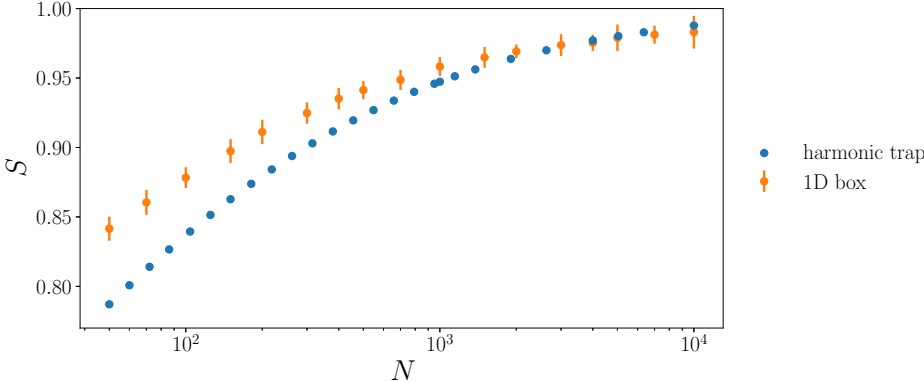

Figure 5: Ratio $S$ between the peak fluctuations in a microcanonical and a canonical ensemble calculation as a function of atom number for a 1D harmonic trap and a 1D box with periodic boundary conditions. The results for the 1D harmonic trap are obtained from an exact calculation (see text). In the case of the 1D box potential, the FSS method was employed.

interactions. Once again, the microcanonical fluctuations are also reduced with respect to the canonical result. Note, that there is no exact calculation which could serve as a benchmark for this case, since to our knowledge, there exists no other method for the weakly interacting gas in the microcanonical ensemble.

### 4.3 Comparison of canonical and microcanonical fluctuations in 1D

Equipped with the analysis above, it is possible to address the natural question, whether the fluctuations in a 1D non-interacting gas depend on the choice of the statistical ensemble in the limit of large atom number $N \to \infty$. To this end we first compare the results for the canonical and the microcanonical ensemble using the exact results for the 1D harmonic oscillator. They allow us to evaluate the ratio $S$ of the peak variance in both ensembles as introduced in Eq. (14). Figure 5 (blue points) shows the result, where $S$ tends to 1, indicating that the microcanonical and canonical fluctuations in the 1D harmonic oscillator asymptotically become equal, in agreement with previous discussions [28].

We also study the same problem for non-interacting atoms in a 1D box potential with periodic boundary conditions as discussed in Sec. 4.1. In this case, no method of finding the exact values of the fluctuations in the microcanonical ensemble is available. Instead, we use our numerical FSS method analysis to obtain the ratio $S$ as shown in Fig. 5 (orange points). Again, we observe that $S$ tends to 1, indicating that the microcanonical and canonical fluctuations in the 1D box potential become equal in the limit of large atom number.

This finding supports the typical notion that results in the thermodynamic limit should be independent of the thermodynamic ensemble. Note however, that this equivalence between the ensembles is not universal, as pointed out previously [9, 28]. In particular we show in the following section, devoted to the experimentally relevant 3D case, that the microcanonical and canonical fluctuations differ even in the limit of large atom numbers.

## 5 Fluctuations in 3D systems

Despite its fundamental importance, the question of atom number fluctuations in Bose-Einstein condensates [28] remained largely an academic problem until a few years ago. Recently how-

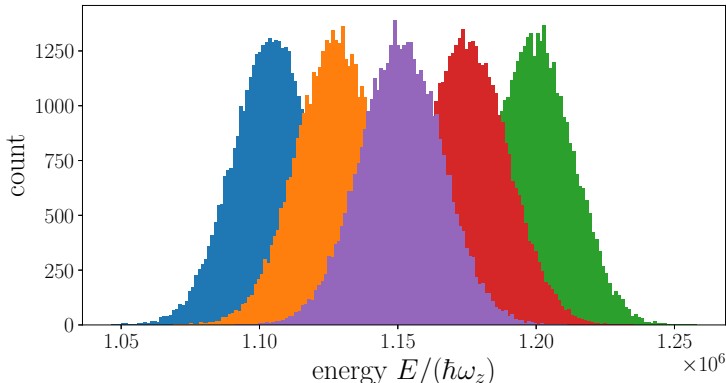

Figure 6: Histograms of the distribution of energies of sets of states generated with the FSS method. The sets represent $N = 10^4$ atoms in a 3D harmonic trap with aspect ratio $\lambda = 4$ at temperatures, from left (blue) to right (green), $k_\mathrm{B} T / (\hbar \omega_z) = 47.5, 47.75, 48, 48.25, 48.5$.

ever, the situation has changed, since improved control of the total number of atoms in ultracold gases has allowed for first measurements of the fluctuations [17, 18].

In the limit of very large atom numbers in a 3D system the asymptotic atom number fluctuations in a microcanonical ensemble calculation [9] are given by

$$\lim_{N \to \infty} \frac{\left( \Delta^2 N_0 \right)_{\mathrm{micro}}}{N} = \left( \frac{\zeta(2)}{\zeta(3)} - \frac{3}{4} \frac{\zeta(3)}{\zeta(4)} \right) \approx 0.53 \,, \tag{18}$$

where $\zeta$ denotes the Riemann zeta function. However, in the following we show that despite the large number of up to $10^5$ atoms, the asymptotic value in Eq. (18) is still not applicable. On the other hand, these experimentally relevant atom numbers are so large that one cannot compute the expected microcanonical fluctuations using previously existing methods, even in the non-interacting case. Here we show how this problem can be solved for large atom numbers by computing the relevant microcanonical fluctuations with the FSS method after appropriate post-selection.

As outlined above, we first generate a set of states representing the non-interacting gas at thermal equilibrium in the canonical ensemble. The atom number variance in such a set is compared with the results obtained from the recurrence relations [5], which ensures that the resulting set is indeed a good representation of the canonical ensemble.

The set of states representing the system at a given temperature has a distribution of energies dictated by statistics in the canonical ensemble. Figure 6 shows histograms of these energies for $N = 10^4$ atoms in a 3D harmonic trap with aspect ratio $\lambda = 4$ at various temperatures.

To obtain a set of states representative to the microcanonical ensemble we perform a post-selection analysis also outlined above. In practice, we post-select a set of states at a given temperature to reduce the variance of energies by subsequential removal of states at the highest and lowest energies, symmetrically with respect to the mean energy. The final set thus has an energy in the interval $[E_{\mathrm{mean}} - \Delta E/2, E_{\mathrm{mean}} + \Delta E/2]$, where $E_{\mathrm{mean}}$ is the mean energy and the energy window is $\Delta E$. By reducing the energy interval, the microcanonical ensemble, i.e. the limit $\Delta E \to 0$ is approached.

The resulting variance of the condensate atom number is shown in Fig. 7, for $N = 10^4$ atoms in a 3D harmonic trap with different aspect ratios at the temperatures $T_{\mathrm{max}}$ for which the canonical fluctuations are maximal. The variances are presented as a function of the fraction $f$, which is defined as the fraction of the number of remaining states with respect to the

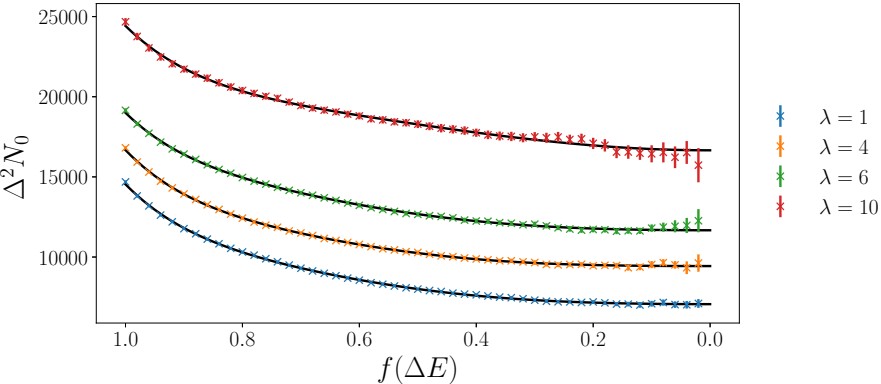

Figure 7: Fluctuations of atom number obtained after post-selection. The atom number variance $\Delta^2 N_0$ for different aspect ratios $\lambda$ is shown as a function of the fraction $f$ after post-selection. Thus the figure represents the transition from a canonical to a microcanonical result from left to right with solid black lines showing the polynomial fits. The initial sets represent $N = 10^4$ atoms at the temperature $T_{\max}$ at which the canonical variance is maximal.

initial states. We repeat our analysis several times, starting from different sets of states in the canonical ensemble and the error bars in Fig. 7 correspond to our statistical uncertainty. For significant post-selection and small resulting sets of states (small $f$) the uncertainty of our result can become significant. Therefore a polynomial fit is used to find the asymptotic value of the variance for $\Delta E \to 0$, which corresponds to the size of the fluctuations in a microcanonical ensemble.

Note that experiments typically suffer from atom loss and technical heating. Moreover, the experimental results are extracted from multiple experimental realizations by using a correlation technique [17, 18]. Hence, even in the absence of interactions the experiment would not correspond to this ideal $\Delta E \to 0$ limit and we therefore expect the measured atom number variance in the Bose-Einstein condensate to lie somewhere between its extremal values, i.e. between the microcanonical and canonical fluctuations.

Figure 7 shows that the atom number variance $\Delta^2 N_0$ is significantly reduced in the transition from the canonical ($f = 1$) to the microcanonical ensemble (limit $f \to 0$). While the results are qualitatively similar, the quantitative reduction depends on the aspect ratio of the trapping potential $\lambda$. Importantly, the fluctuations differ significantly in a canonical and a microcanonical calculation even in a very elongated trap ($\lambda = 10$) for $N = 10^4$ atoms.

## 5.1 Comparison of canonical and microcanonical fluctuations in 3D

Based on the analysis in a 3D harmonic trap, once again the question arises whether the fluctuations depend on the choice of the statistical ensemble in the limit of large atom number $N \to \infty$. We study this problem for the non-interacting 3D harmonically trapped gas by evaluating the ratio $S$ between microcanonical and canonical fluctuations for experimentally relevant aspect ratios $\lambda$ from 1 to 20 and up to $10^5$ atoms.

Figure 8 shows the ratio between microcanonical and the canonical fluctuations evaluated with an FSS method analysis and using exact recurrence relations for small atom numbers. Note that the FSS method does not provide the ratio $S$ directly and we therefore take the following approach. In the analysis the set of states representing the canonical ensemble for which $\Delta^2 N_{0\text{cano}}$ is maximal is reduced by post-selection and converges to a set consistent with the microcanonical ensemble. However, this is not necessarily the set for which the fluctuations $\Delta^2 N_{0\text{micro}}$ reach their maximum and we therefore denote the resulting ratio of the fluctuations

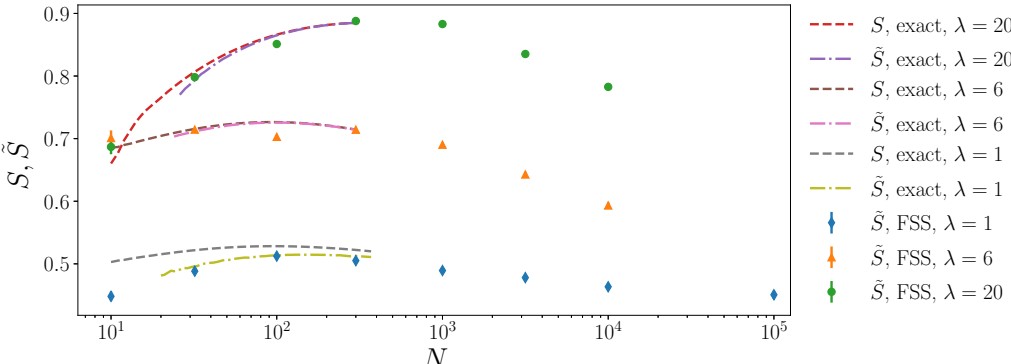

Figure 8: Ratio between microcanonical and the canonical fluctuations in a non-interacting harmonically trapped 3D gas. The coefficients $S$ (the ratio of the peak fluctuations in both ensembles, see eq. 14) and $\tilde{S}$ (see text) characterizing this ratio are shown as a function of the number of atoms $N$ for different aspect ratios. Data points indicate solutions of the FSS method analysis. Solid lines represent exact results obtained using the recurrence relations given in Appendix B.

by $\tilde{S}$. It may differ from $S$ due to two reasons. Firstly, the microcanonical fluctuations should be labelled by the proper control parameter, i.e. energy. However, in the FSS method we assume that the temperature is inherited from the canonical ensemble. Secondly, for a small number of atoms, the maximal fluctuations are reached at slightly different temperatures in both ensembles. To evaluate the effect, we compute $\tilde{S}$ and $S$ using an exact method (see Appendix B) as shown in Fig. 8. This shows that the two ratios agree for atom numbers above a few hundred atoms and validates the use of $\tilde{S}$ obtained from our FSS method analysis for larger atom numbers.

Overall, Fig. 8 shows that the ratio between microcanonical and the canonical fluctuations first grows and then starts to decrease. This growth for small number of atoms is easily understood. In an elongated trap, the atoms easily populate the low lying energy levels associated with the longitudinal direction and thus the system becomes effectively one-dimensional, discussed in the previous section. For $N$ sufficiently large, however, the 3D character of the trap becomes relevant, since the levels in the transverse directions also become populated. This is in striking contradiction with the result in the strict $1D$ case where $S$ approaches 1 (compare with Fig. 5) and clearly breaks with the notion that results in the large atom limit should be independent of the thermodynamic ensemble.

Note that the expected value of $S$ in the limit $N \to \infty$ is given by

$$S_{3D} = \left(1 - \frac{3\zeta(3)^2}{4\zeta(4)\zeta(2)}\right) \approx 0.39, \tag{19}$$

which is the ratio between the asymptotically known microcanonical fluctuations, Eq. (18), and fluctuations in the canonical ensemble [7]

$$\lim_{N \to \infty} \frac{\left(\Delta^2 N_0\right)_{\text{cano}}}{N} = \frac{\zeta(2)}{\zeta(3)}. \tag{20}$$

The FSS method analysis for $10^5$ atoms in a spherical trap results in $S \approx 0.45$ which approaches the asymptotic value 0.39.

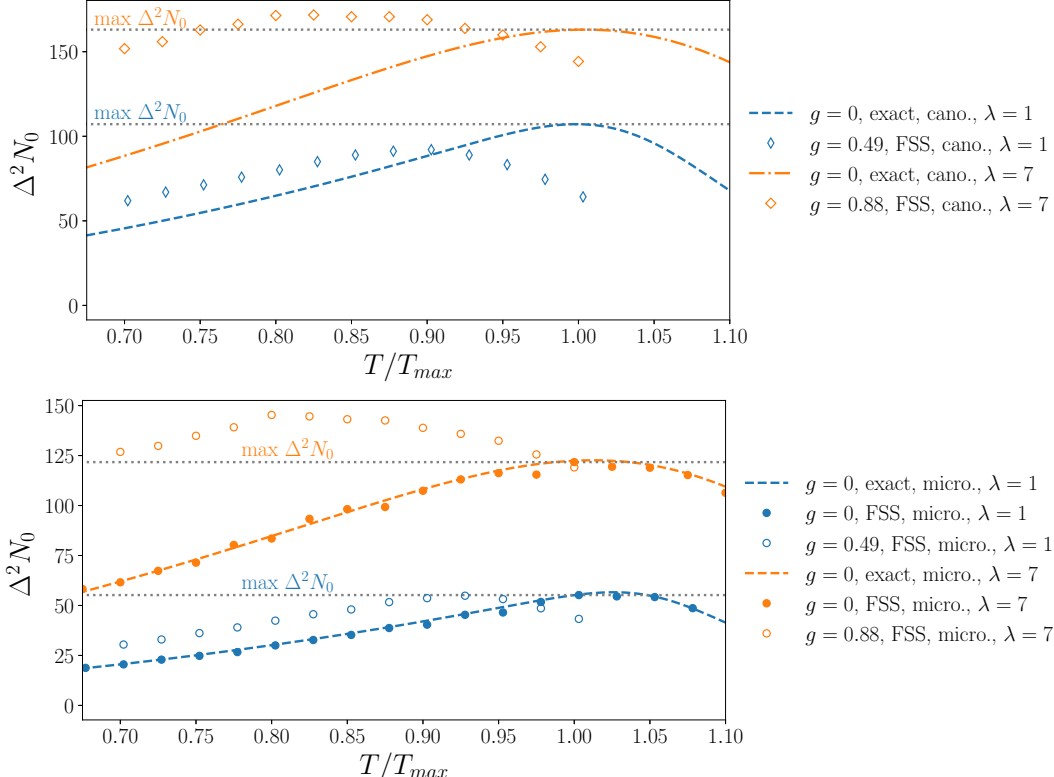

Figure 9: Atom number fluctuations with and without interactions in a 3D harmonically trapped gas containing $N = 100$ atoms, computed in the canonical (top) and microcanonical (bottom) ensemble. The variance of the atom number is shown as a function of temperature for a spherically symmetric $\lambda = 1$, (blue) and an elongated $\lambda = 7$ (orange) trap with and without interactions. The horizontal dashed lines correspond to the maximal condensate atom number fluctuations for the non-interacting gas in each case. The reference temperature $T_{\max}$ is the temperature of maximal fluctuations of the non-interacting gas in the canonical ensemble. The units of the interaction strength $g$ are $(m\omega_z)^{\frac{3}{2}}\omega_\perp/\sqrt{\hbar}$.

## 5.2 Microcanonical fluctuations in an interacting 3D gas

Finally, the Fock state sampling method allows for the investigation of weak repulsive interactions in a 3D harmonically trapped gas, which remains a controversial problem. In particular, different theoretical approaches do not even agree whether interactions lead to an increase or a decrease of fluctuations (as summarized in Fig. 4 of [17]). In the case of 1D confinement this was discussed in Sec. 4. There it was shown that the direction of the shift due to interactions is temperature dependent and thus theories valid at different temperatures come to different conclusions. Moreover, the direction of the shift depends on the particular system e.g. at low temperatures interactions lead to decrease of fluctuation in the 1D ring trap, while an increase is observed for the 1D harmonic potential.

In this sense, it is of particular importance to evaluate the interactions in the experimentally relevant 3D harmonic potential. The FSS method allows for such an analysis for very weak interactions where the state of the Bose-Einstein condensate does not differ significantly from the lowest harmonic oscillator state. Moreover, the post-selection process introduced above allows for the calculation of the fluctuations in the microcanonical ensemble.

Figure 9 shows the atom number fluctuations for a non-interacting and interacting gas, in

the canonical (upper panel) and the microcanonical (lower panel) ensembles. A comparison between the upper and lower panel shows the clear overall suppression of fluctuations in the microcanonical case as discussed in Sec. 5.1. Moreover, each panel shows that the fluctuations in a spherical trap are generally lower than the ones observed in an elongated geometry in accordance with the previous findings of Fig. 8.

To discuss the temperature dependence, it is normalized to the temperature of maximal fluctuations of the non-interacting gas in the canonical ensemble $T_{\max}$. This avoids an ambiguity in the definition of the critical temperature for finite systems. In the canonical case the interactions lead to a shift of the peak fluctuations to lower temperatures as expected for a shift of the critical temperature due to interactions. This is reminiscent of the effect observed in the 1D harmonic configuration of Fig. 4. The shift leads to increased fluctuations at low temperature and a decrease at $T_{\max}$. Importantly, the value of the peak fluctuations depends on the aspect ratio $\lambda$ and can both increase or decrease as shown in Fig. 9 (upper panel).

Finally, the experimentally most relevant case are the fluctuations in the microcanonical ensembles as shown in Fig. 9 (lower panel). Similar to the previous case interactions lead to a shift of the peak fluctuations and a resulting increase at low temperature. The peak fluctuations do not show a uniform behaviour as they increase in an elongated trap but remain constant in a spherical geometry.

Our results clearly show that the effect of interactions on the atom number fluctuations in a Bose-Einstein condensate depend both on the temperature and the trapping geometry of the gas. In that sense one can not expect a single answer to the question whether interactions increase or decrease the fluctuations in a Bose-Einstein condensate.

# 6 Conclusion

In this paper the fluctuations of a Bose-Einstein condensate were investigated in the non-interacting gas and in the case of very weak interactions. Using the Fock state sampling method we studied the fluctuations in different trap geometries, and in the canonical and microcanonical ensembles as a function of temperature.

In a 1D system with periodic boundary conditions our results agree with the classic result of S. Giorgini, L. Pitevskii and S. Stringari [19] based on the Bogoliubov approximation for low temperatures. However, the fluctuations increase for higher temperatures and the maximal fluctuations are indeed larger than to the non-interacting result. Thus, we resolved a long standing controversy by showing that the Bogoliubov approximation only leads to reliable results at low temperatures.

For the case of the 1D harmonically trapped gas we showed that interactions lead to a clear shift of the peak fluctuations to lower temperatures. This is in general agreement with the expected shift of the critical temperature and leads to increased fluctuations at low temperatures. Moreover, we employed a post-selection process to the FSS method to evaluate the fluctuation in the microcanonical ensemble. This showed a clear reduction of the fluctuations with respect to the canonical expectation for a gas containing 100 atoms. Nonetheless, a general analysis showed that the canonical and microcanonical results agree in the limit of large atom numbers in 1D.

Finally, we investigated the experimentally most relevant case of a 3D harmonically trapped gas. In this case the microcanonical calculation yields a reduction of the peak fluctuations which depends on both aspect ratio and atom number. Importantly, our FSS method results for large atom numbers slowly approach the expected asymptotic value and thus confirm that the canonical and microcanonical fluctuations do not agree in the limit of large atom numbers in 3D.

The weakly interacting 3D harmonically trapped gas shows fluctuations which are both shifted in temperature and altered in amplitude depending on temperature and trapping geometry. Thus it is clearly not possible to provide a universal rule for the effect of interactions on the fluctuations of a Bose-Einstein condensate.

In view of these results, recent experiments [18] should be compared to calculations in the microcanonical ensemble. In future work we will extend the FSS method to include the realistic condensate wave-function. This will allow us to make quantitative predictions for the fluctuations in realistic experimental systems. Moreover, we will be able to map out the effect of the ensemble choice and the interactions in a large parameter space.

# Acknowledgements

We thank P. Deuar for fruitful discussions and Laurits Nikolaj Stokholm for valuable comments on the manuscript.

**Funding information** M. B. K. acknowledges support from the (Polish) National Science Center Grant No. 2018/31/B/ST2/01871. K. P. acknowledges support from the (Polish) National Science Center Grant No. 2019/34/E/ST2/00289. K. R. and M. B. K. acknowledge support from the (Polish) National Science Center Grant No. 2021/43/B/ST2/01426. This research was supported in part by PLGrid Infrastructure. Center for Theoretical Physics of the Polish Academy of Sciences is a member of the National Laboratory of Atomic, Molecular and Optical Physics (KL FAMO). This work has been supported by the Danish National Research Foundation through the Center of Excellence "CCQ" (Grant agreement No. DNRF156) and by the Independent Research Fund Denmark - Natural Sciences via Grant No. 8021-00233B and 0135-00205B.

# A   Technical details of the algorithm

The FSS method is a realization of the Metropolis algorithm that samples multimode Fock state configurations in the canonical ensemble. Let $|\theta\rangle = |N_0, N_1, \ldots\rangle$, $|\theta'\rangle = |N_0', N_1', \ldots\rangle$ be the Fock states representing respectively the initial state and a slightly modified candidate. In the FSS method, we restrict ourselves in generating $\theta'$ to only the ones that amount to moving a single particle from one orbital $\phi_i$ to another $\phi_j$, compared to the original state $\theta$, that is $N_i' = N_i - 1$, $N_j' = N_j + 1$ and $N_k' = N_k$ for $k \neq i \wedge k \neq j$. Let $q_A(\theta, i)$ and $q_B(\theta, j)$ be the probabilities of randomly selecting orbitals $\phi_i$ and $\phi_j$ (note the explicit dependence on the state $\theta$). We define the proposal distribution

$$Q(\theta'|\theta) = q_A(\theta, i) q_B(\theta, j), \tag{21}$$

with

$$q_A(i) \propto \exp(-\gamma E_i) N_i, \quad q_B(j) \propto \exp(-\gamma E_j)(N_j + 1), \tag{22}$$

where $\gamma > 0$ is a parameter of the method and $E_i$, $E_j$ are the single particle energies of their respective orbitals. The exponential factors are introduced to remedy the wasteful jumps in high energy orbitals. The parameter $\gamma$ allows for tuning of the acceptance rate of the algorithm and thus optimizing its convergence rate. The proposal distribution is not symmetric for $\gamma \neq 0$, that is $Q(\theta'|\theta) \neq Q(\theta|\theta')$ and it is taken into account, however the ratio $Q(\theta'|\theta)/Q(\theta|\theta') \approx 1$ when the number of particles is large (on the order of 100 and above). The tuning parameter $\gamma$ takes for example values of about 0.2 and 0.1 for 100 and 1000 particles, respectively, in a spherical harmonic trap.

In our implementation of the algorithm, the complexity of calculating the energy difference between states $\theta$ and $\theta'$ is $O(\log^2 N)$ in the non-interacting case and $O(N^2)$ in the interacting case, where $N$ is the total number of particles. The algorithm scales perfectly for large computer clusters or CPUs with large number of cores as multiple independent instances can be launched simultaneously and after thermalization, each instance produces independent samples from the same ensemble.

# B  Recurrence relations used to compute the partition functions

## B.1  Recurrences for microcanonical partition function in a harmonic trap

Following [34] we introduce, a new function of three arguments, $\tilde{\Gamma}_{\text{ex}}(N, E, \epsilon)$. This function is the number of ways to distribute $N$ particles between excited energy levels such that the total energy is $E$, but with the restriction that energy levels above $\epsilon$ are empty.

With this definition we have the relation

$$\tilde{\Gamma}_{\text{ex}}(N, E, \epsilon = 0) = \delta_{N,0}\,\delta_{E,0}\,, \tag{23}$$

where, as before, we assume that the the energy levels are written in oscillator units and are therefore given by integers.

The values of $\tilde{\Gamma}_{\text{ex}}(N, E, \epsilon)$ for other energy thresholds $\epsilon$ follow the recurrence

$$\tilde{\Gamma}_{\text{ex}}(N, E, \epsilon_j) = \sum_{n_j=0}^{N} \tilde{\Gamma}_{\text{ex}}(N - n_j, E - n_j \epsilon_j, \epsilon_j - 1) \binom{\mathcal{D}(\epsilon_j) + n_j - 1}{\mathcal{D}(\epsilon_j) - 1}, \tag{24}$$

where $\mathcal{D}(\epsilon_j)$ is a degeneracy of the energy level $\epsilon_j$.

The label $n_j$ in the recurrence (24) has the meaning of the number of atoms occupying the energy level $\epsilon_j$. There are $\binom{\mathcal{D}(\epsilon_j) + n_j - 1}{\mathcal{D}(\epsilon_j) - 1}$ ways of distributing $n_j$ indistinguishable bosons between $\mathcal{D}(\epsilon_j)$ levels. If exactly $n_j$ atoms seats in the energy level $\epsilon_j$, then the remaining $N - n_j$ ones occupy energy levels up to $\epsilon_j - 1$, such that their total energy is $E - n_j \epsilon_j$.

The microcanonical partition function for the excited atoms, i.e. $\Gamma_{\text{ex}}(N_{\text{ex}}, E)$, can be obtained from the auxiliary function $\tilde{\Gamma}_{\text{ex}}(N, E, \epsilon)$ using the relation

$$\Gamma_{\text{ex}}(N, E) = \tilde{\Gamma}_{\text{ex}}(N, E, E)\,, \tag{25}$$

which expresses the fact that there is no partition leading to the total energy $E$ which would involve energy levels higher than $E$.

As described in the main text, the microcanonical partition function is given by

$$\Gamma(N, E) = \sum_{N_{\text{ex}}=0}^{N} \Gamma_{\text{ex}}(N_{\text{ex}}, E)\,, \tag{26}$$

linebreak and the probability of finding $N_0$ atoms in the ground state is

$$p_{\text{micro}}(N_0, N, E) = \Gamma_{\text{ex}}(N - N_0, E)/\Gamma(N, E)\,. \tag{27}$$

Alternatively, one can directly use a recurrence for an auxiliary partition function $\tilde{\Gamma}(N, E, k)$, defined similarly to the previous one, but including atoms in the ground state. It still obeys the recurrence relation (24), but with a slightly different initial condition $\tilde{\Gamma}(N, E, k = 0) = \delta_{E,0}$. In this implementation the probability of finding exactly $N_{\text{ex}}$ excited atoms is

$$p_{\text{micro}}(N_0, N, E) = \frac{1}{\tilde{\Gamma}(N, E, E)} \left( \tilde{\Gamma}(N - N_{\text{ex}}, E, E) - \tilde{\Gamma}(N - N_{\text{ex}} - 1, E, E) \right). \tag{28}$$

There are other recurrence relations, see for instance [5], but (24) turned out easy to program and relatively efficient.

## B.2 Recurrence for canonical partition function in a — harmonic trap

To compute the canonical partition function $Z(N, \beta)$ we invoke the recurrence relation

$$Z(N, \beta) = \frac{1}{N} \sum_{n=1}^{N} Z(1, n\beta) Z(N - n, \beta). \tag{29}$$

This relation appears in many contexts and is nicely described in [5].

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
