# Peer review of "Microcanonical and Canonical Fluctuations in atomic Bose-Einstein Condensates -- Fock state sampling approach"

_SciPost Physics, doi:SciPost Phys. 14, 036 (2023)_

## Round 1 · Referee Report · Anonymous · 2022-6-26

Strengths

1) Develpment of Fock state sampling method
2) Comparison of micro-canonical and canonical fluctuation results for experimentally realistic situations

Weaknesses

see the report below

Report

The concisely written manuscript studies the fluctuations of a Bose-Einstein condensate for vanishing and weak interaction in different trap geometries for both the canonical and the micro-canonical ensemble. Due to the introduction of the Fock sampling method impressive results are obtained, which compare well with the literature on the one hand and which allow to tackle larger particle numbers as well as higher temperatures in comparison with previous methods on the other hand. The latter progress is important in view of recent experiments, where the atom number fluctuations were measured so precisely that now a quantitative comparison with theoretical predictions is needed.

The high quality of the manuscript deserves publication in SciPost. However, prior to pubication the authors should address the requested changes.

Requested changes

1) Caption of Fig. 1 is unprecise:

a) Is a 3D non-interacting Bose gas considered?

b) Is the homogeneous or the harmonically trapped case considered?

2) Ref. [13] reports measurements for a photon BEC, which reveal a crossover from a canonical to a grand-canonical behavior. This leads to the question how one could interpolate in Fig. 1 between the fluctuations of the number of ground state atoms between the three distinct statistical ensembles. In this respect it is unclear to me whether such an interpolation would be possible in a unique way or whether, in principle, different interpolation schemes might occur. The authors should comment on this.

3) Page 4: the the

  • validity: top
  • significance: top
  • originality: top
  • clarity: top
  • formatting: perfect
  • grammar: perfect

Author:  Maciej Kruk  on 2022-09-08  [id 2794]

(in reply to Report 1 on 2022-06-26)
Category:
answer to question
correction

We thank the Referee for careful revision of the manuscript and for the encouraging comments on the high quality of the manuscript.

Here we address her/his comments.

1) Caption of Fig. 1 is unprecise:

a) Is a 3D non-interacting Bose gas considered?

b) Is the homogeneous or the harmonically trapped case considered?

The sketch was meant to be a graphical introduction to the topic, and therefore it was missing details. Following the referee's advice we decided to make it more precise. We added to the caption the sentence: "This sketch is based on a numerical calculation for \(N=1000\) non-interacting atoms in a three dimensional symmetric harmonic trap."

We also added in the text: "Figure~1 is a sketch that outlines the expected atom number fluctuations of a a non-interacting Bose gas calculated using the statistical ensembles discussed above for the case of \(N=1000\) atoms trapped in a 3D harmonic trap. This clearly shows that the expected fluctuations strongly depend on the chosen ensemble."

2) Ref. [13] reports measurements for a photon BEC, which reveal a crossover from a canonical to a grand-canonical behavior. This leads to the question how one could interpolate in Fig. 1 between the fluctuations of the number of ground state atoms between the three distinct statistical ensembles. In this respect it is unclear to me whether such an interpolation would be possible in a unique way or whether, in principle, different interpolation schemes might occur. The authors should comment on this.

This is an interesting question. In fact, we present a route from canonical to microcanonical fluctuations in Fig. 7. While the endpoints, canonical and microcanonical results are unique, the path linking them is not. In our case, we perform a post-selection of the set of points representing the canonical ensemble by restricting the set of points to a sharp shrinking interval of energies around the most probable value. To the same end, the microcanonical result could be obtained by a post-selection with a smooth, shrinking filter function such as e.g. a Gaussian function.

The transition between grand canonical and canonical ensembles is a more problematic problem. To this end, we would need to generalize our Fock States Sampling method to generate the approximate grand canonical ensemble. Although we have some ideas of how to build a suitable generalization, this is a new and certainly an interesting project that remains outside of the scope of this paper. Even if done, again the route from the grand canonical fluctuations to the canonical ones would also be non-unique.

The problem of condensate fluctuations in a photonic condensate is studied in Ref. 13. Here a model of coupling to a large, but finite size reservoir should be constructed. Again, although interesting, it certainly remains outside of the scope of this paper.

In response we have added the following sentences: "Importantly, the post-selection process allows for an investigation of the transition from the canonical to microcanonical ensembles. However, it does not give access to the transition from grand canonical to canonical ensembles, which is of interest for photonic condensates [13] and will be the topic of future research."

3) Page 4: the the

We have corrected the typo.

---

## Round 1 · Referee Report · Anonymous · 2022-7-27

Strengths

1) The paper contributes a new theoretical method to model fluctuations in Bose condensates, an old topic that has recently gained momentum and renewed interest in quantum gases, following new experimental developments.

2) The results contribute to our understanding of the fluctuation behavior of the Bose gas in the presence of interactions and different trap geometries in 1D and 3D.

Weaknesses

1) The generic title “Microcanonical and canonical fluctuations in Bose-Einstein Condensates” begs for a discussion of the connection to Bose gases of photons, where canonical and grand canonical fluctuations have been observed (Ref.13). This link is presently missing and should be elaborated more if the paper aims to be of broader interest, beyond the cold-atom system from the Aarhus group.

2) The studied systems settings in the present manuscript seem random (1D: box vs. harmonic potential, canonical vs. microcanonical ensemble; 3D: canonical vs. microcanonical (harmonic), noninteracting vs. interacting (microcanonical)). What about 2D? What about a 3D box? The authors should justify their choice of specific system settings.

Report

The authors present a theoretical study of the number fluctuations in a Bose-Einstein condensate, where particles are allowed to transition between different eigenstates of the system. The used numerical method is a Metropolis algorithm, which allows for the extraction of different particle number configurations and, accordingly, the number fluctuations. In their calculations, the authors probe the strength of the fluctuations as a function of the temperature for different potential geometries (flat, harmonic), dimensions (1D, 3D), interactions (noninteracting, weakly repulsive) in the canonical and microcanonical ensemble.

The main findings are that fluctuations are typically more suppressed in the microcanonical ensemble with respect to the canonical one, especially at small total particle numbers. In the presence of weak interactions, the fluctuations as a function of temperature are observed to be modified. In 3D, the microcanonical ensemble shows less fluctuations compared to the canonical one, while for the latter the authors investigate the fluctuations as a function of trap aspect ratio and temperature. As in 1D, interactions are observed to modify the fluctuations in a nonuniversal way.

Overall, this is an important, well-written paper, with plausible results. The demonstrated methods enable a modelling of fluctuating Bose condensates seen in recent experiments. At several points physical explanations should be added to clarify the findings of the paper (see list below); in particular, the authors should better motivate their choice of system configurations (see comment above).

I recommend publication of the manuscript after addressing these points, and the ones (mostly typos) listed below.

Requested changes

- Section 1: “The questions was further” –> “question”

- Fig. 1: The quantity shown on the y-axis is not introduced during or before the discussion of the figure. According to eq. (3), this should be variance of the BEC population. This should be clarified in Section 1, for example.

- Fig. 1: Explain why the maxima of the orange and blue graphs are not at Tc? The caption just says they can be shifted relative to each other. What is the total particle number N? Since absolute numbers are given on the y-axis, N should be stated explicitly for comparison.

- Section 2: After eq. (1), the energy density operator \hat h is introduced. What is \Delta?

- Section 2: Typo “solely on the the parameter”

- Section 2: “where \hat \rho is the density matrix of gas at equilibrium” –> “the gas”

- Section 3: “In practice, we also truncate the orbitals at some high value, but now the results are cut-off independent, provided that the cut-off is suffciently high.” What is the criterion for “sufficiently”, some ratio E_j/k_B T > \eta?

- Section 3: “we only consider the statistics of the population in the lowest orbital in the present version of the method, even in the interacting case” I would appreciate a clarifying sentence, what would change if a Thomas Fermi profile was used instead of the Gaussian ground state.

- Section 3: “W is the number of states”, the term ‘states’ is easily confused with the states of in the box/harmonic trap. Rather call it ‘configurations’ or ‘copies’ or ‘samples’…

- Section 3: “Figure 1 illustrates these peak fluctuation in both ensembles.” –> “fluctuations”

- Fig. 2: The physical meaning of the temperature of the maximum should be discussed. In the text the authors only write that it agrees with the other models, but do not describe what is shown in the plot.

- Fig. 3, inset: add axes labels; also indicate with vertical lines at T=5 and T_max~=23 that the inset analyses these particular temperatures.
In the text the authors should mention why – from a physics perspective – the fluctuations are altered by the interactions: Is it due to a different energy cost?

- In the text it says “At higher temperatures the effect reverses and the weakly interacting microcanonical variance is larger” which is not shown anywhere. Does this occur at even larger T?

- Fig. 4: “Fluctuations of Bose gas…” –> “Fluctuation of the ground state population in a Bose gas containing N=100…”

- Fig. 4: It is very difficult to read the caption and see the connection with the many lines and symbols. Add these descriptions, e.g. FSS method (blue symbols), Bogoliubov (orange dotted, dashed lines) …

- Fig. 5: The ratio S is studied as a function of particle number N up to 10^4, which is close to experimentally relevant numbers. It is not clear to me why in the previous plots, the authors show data for N=100 and not for larger ensembles. A clarification would be appreciated

- Eq. (18): what is the numerical value of this limit, and why not show \Delta^2 N_0 / N^2?

- Fig. 7: Describe solid lines

- Fig. 8: In the caption explicitly say what S is, i.e., the ratio of the ‘maxima’ of the fluctuations.

- Section 5.1: “Note that the expected asymptotic value of S is given by”. Please say again asymptotic with respect to which quantity? \lambda? One could show S=0.39 as a horizontal line in the plot of Fig. 8 for better understanding.

- Section 5.2.: “[…] whether interactions lead to an increase […] of interactions”. The second ‘interactions’ should be replaced by ‘fluctuations’

  • validity: high
  • significance: good
  • originality: high
  • clarity: good
  • formatting: good
  • grammar: excellent

Author:  Maciej Kruk  on 2022-09-08  [id 2795]

(in reply to Report 2 on 2022-07-27)
Category:
answer to question
correction

We thank the Referee for very careful reading, valuable comments and positive remarks on our work.

We first answer to the two main comments of the Referee and then address minor remarks.

The Referee commented:

1) The generic title “Microcanonical and canonical fluctuations in Bose-Einstein Condensates” begs for a discussion of the connection to Bose gases of photons, where canonical and grand canonical fluctuations have been observed (Ref.13). This link is presently missing and should be elaborated more if the paper aims to be of broader interest, beyond the cold-atom system from the Aarhus group

In our paper we presented a method to simulate the canonical and microcanonical ensembles, and the routes between them, as indicated by the title. Although we show the method in the context of experiments by the Aarhus group, the method is general.

Reference 13 treats the grand canonical ensemble for photons, and then shows how one can converge towards the canonical ensemble via shrinking a reservoir.

From the computational point of view, the smooth transition between grand canonical and canonical ensembles is more problematic than between the canonical and microcanonical ensembles. To this end, we would need to generalize our Fock States Sampling method to generate the approximate grand canonical ensemble. Although we have some ideas of how to build a suitable generalization, this is a new and certainly an interesting project that remains outside of the scope of this paper. Even if done, the route from the grand canonical fluctuations to the canonical ones would not be unique.

Moreover, in Ref. 13 a model of coupling to a large, but finite size reservoir should be constructed, which is even one step further. Again, although interesting, it certainly remains outside of the scope of this paper.

In response to the referee's comment we have decided to change the title to "Microcanonical and Canonical Fluctuations in atomic Bose-Einstein Condensates - Fock state sampling approach". This clarifies that our work primarily considers atomic systems.

Moreover we have added the following sentences: "Importantly, the post-selection process allows for an investigation of the transition from the canonical to microcanonical ensembles. However, it does not give access to the transition from grand canonical to canonical ensembles, which is of interest for photonic condensates [13] and will be the topic of future research."

2) The studied systems settings in the present manuscript seem random (1D: box vs. harmonic potential, canonical vs. microcanonical ensemble; 3D: canonical vs. microcanonical (harmonic), noninteracting vs. interacting (microcanonical)). What about 2D? What about a 3D box? The authors should justify their choice of specific system settings.

Indeed, our Fock States Sampling method is very general and may be applied to study canonical and microcanonical statistics of ultracold Bose gas in various traps and different dimensions. Our paper presents results in 1D and 3D harmonic traps of various aspect ratios. One-dimensional traps offer explicit solutions at least in the canonical ensemble. This offers several benchmarks for our novel Metropolis algorithm. On the other hand, 3D harmonic traps are relevant for the important Aarhus experiments. We stress that the analysis of 2D and 3D boxes will be the subject of our next publication. Moreover, we stress that the box traps with periodic boundary conditions have the universal, interaction-independent condensate wave function. Hence the present version of our method is particularly well suited to this case.

In response to the referees comment we have added the following paragraph and shortened the outline of the paper accordlingly: "To benchmark our method we first study the Bose gas in a one dimensional box potential with periodic boundary conditions and in a harmonic trap. In these systems, comparisons with exact findings and with the other approximate methods (such as the Bogoliubov and the classical field approximation) are possible. We then proceed to apply our method to the experimentally more relevant three dimensional systems." This clarifies the choice of systems under investigation.

Answers to other remarks

We are grateful to the referee for the very careful reading our manuscript.

Below we address all other remarks of the Referee.

- Fig. 1: The quantity shown on the y-axis is not introduced during or before the discussion of the figure. According to eq. (3), this should be variance of the BEC population. This should be clarified in Section 1, for example.

We have change the y-axis label to "BEC fluctuations", and we modified the caption changing "Illustration of the fluctuations of the number of ground state atom" to "Illustration of the variance of the number of ground state atom".

- Fig. 1: Explain why the maxima of the orange and blue graphs are not at \(T_C\)? The caption just says they can be shifted relative to each other. What is the total particle number N? Since absolute numbers are given on the y-axis, N should be stated explicitly for comparison.

Fig. 1 was just meant to be a sketch, like a graphical abstract. We clarified it in the main text, by adding: "Figure 1 is a sketch that outlines the expected atom number fluctuations of a a non-interacting Bose gas calculated using the statistical ensembles discussed above for the case of \(N=1000\) atoms trapped in a 3D harmonic trap. This clearly shows that the expected fluctuations strongly depend on the chosen ensemble."

- Section 2: After eq. (1), the energy density operator \(\hat{h}\) is introduced. What is \(\Delta\)?

Here \(\Delta\) is just the Laplace operator. We added a hat to mark it.

- Section 2: Typo “solely on the the parameter”

We corrected the typo.

- Section 3: “In practice, we also truncate the orbitals at some high value, but now the results are cut-off independent, provided that the cut-off is suffciently high.” What is the criterion for “sufficiently”, some ratio \(E_j/k_B T > \eta\)?

When performing calculations we gradually increase the cut-off until the results stabilize. This is a fairly common way of determining numerically the limit of a series. The intuition of the Referee is correct: the higher the temperature the higher the cut-off.

- Section 3: “we only consider the statistics of the population in the lowest orbital in the present version of the method, even in the interacting case” I would appreciate a clarifying sentence, what would change if a Thomas Fermi profile was used instead of the Gaussian ground state.

It is the most interesting open question. The Fock State Sampling method, in its present formulation, gives no information about the relative phases between orbitals. We are planning the construction of a more powerful variant of our method that would be formulated based on orbitals that include the correct wave function of the condensate of an interacting gas.

Thus the Thomas-Fermi regime is beyond validity of the current implementation of the method.

- Section 3: “W is the number of states”, the term ‘states’ is easily confused with the states of in the box/harmonic trap. Rather call it ‘configurations’ or ‘copies’ or ‘samples’…

We changed this sentence to "W is the number of configurations, represented in what follows, by Fock states".

- Fig. 2: The physical meaning of the temperature of the maximum should be discussed. In the text the authors only write that it agrees with the other models, but do not describe what is shown in the plot.

The point of maximal fluctuations attracted our attention since it is relatively precisely determined in the recent experiments in Aarhus. As shown in Z. Idziaszek and K. Rzążewski Phys. Rev. A 68, 035604 (2003) the temperature of maximal fluctuations tends to the critical temperature as \(N\) tends to infinity.

- Fig. 3, inset: add axes labels; also indicate with vertical lines at \(T=5\) and \(T_{\rm max}\sim=23\) that the inset analyses these particular temperatures. In the text the authors should mention why – from a physics perspective – the fluctuations are altered by the interactions: Is it due to a different energy cost?

Indeed. The spectrum of the Hamiltonian is modified by interaction, hence the probabilities of jumping in the Metropolis random walk are also modified.

- In the text it says “At higher temperatures the effect reverses and the weakly interacting microcanonical variance is larger” which is not shown anywhere. Does this occur at even larger T?

This part refers to Figs. 2 and 3, where the effect can be seen by comparing the green symbols in both figures. The black line is identical in both figures and serves as a guide to the eye for this purpose. We agree that this is not immediately obvious and the have therefore improved the text as follows: "At higher temperatures the effect reverses and the weakly-interacting microcanonical variance is larger than non-interacting one. This is evident from a comparison of the green symbols in Fig. 2 and Fig. 3, where the non-interacting canonical result can serve as a guide to the eye."

- Fig. 4: It is very difficult to read the caption and see the connection with the many lines and symbols. Add these descriptions, e.g. FSS method (blue symbols), Bogoliubov (orange dotted, dashed lines) …

We improved the caption and we hope in this way Fig. 4 is more readable.

- Fig. 5: The ratio S is studied as a function of particle number N up to \(10^4\), which is close to experimentally relevant numbers. It is not clear to me why in the previous plots, the authors show data for \(N=100\) and not for larger ensembles. A clarification would be appreciated

The calculation for \(10^4\) is already quite involved, especially for the microcanonical ensemble, whereas a simulation for \(100\) is a matter of seconds. Doing computations for \(10^4\) atoms would take months, not adding many information.

- Eq. (18): what is the numerical value of this limit, and why not show \(\Delta^2 N_0 / N^2\)?

We have added the numerical value to the formula that states \(\Delta^2 N_0 / N \approx 0.53\).

- Fig. 7: Describe solid lines

We described the solid lines (results of fitting) in the caption of Fig. 7.

- Fig. 8: In the caption explicitly say what S is, i.e., the ratio of the "maxima" of the fluctuations.

We modified the caption accordingly.

- Section 5.1: “Note that the expected asymptotic value of S is given by”. Please say again asymptotic with respect to which quantity? \(\lambda\)? One could show \(S=0.39\) as a horizontal line in the plot of Fig. 8 for better understanding.

We modified this part by adding "in the limit \(N\to\infty\)". Please note, that the limit \(\lambda\to\infty\) is discussed also in the text - in this limit, but for fixed \(N\), the system becomes 1D.

- Section 5.2.: “[…] whether interactions lead to an increase […] of interactions”. The second ‘interactions’ should be replaced by ‘fluctuations’

We corrected the mistake.

---

## Round 2 · Referee Report · Anonymous (Referee 2) · 2022-9-24

Report

I would like to thank the Authors for their response to my questions. In the revised manuscript, they have fully addressed my remarks, so that I recommend publication of the manuscript in its current form in SciPost Physics.

---

## Editorial Decision

published